# Additive global cerebral blood flow normalization in arterial spin labeling perfusion imaging

Stephanie B. Stewart[1,2], Jonathan M. Koller[2], Meghan C. Campbell[1,3], Joel S. Perlmutter[1,3,4,5,6] and Kevin J. Black[1,2,3,4,5]

[1] Department of Neurology, Washington University School of Medicine, St Louis, MO, USA
[2] Department of Psychiatry, Washington University School of Medicine, St Louis, MO, USA
[3] Department of Radiology, Washington University School of Medicine, St Louis, MO, USA
[4] Departments of Anatomy and Neurobiology, Washington University School of Medicine, St Louis, MO, USA
[5] Division of Biology and Biomedical Sciences, Washington University School of Medicine, St Louis, MO, USA
[6] Programs in Physical Therapy and Occupational Therapy, Washington University School of Medicine, St Louis, MO, USA

## ABSTRACT

To determine how different methods of normalizing for global cerebral blood flow (gCBF) affect image quality and sensitivity to cortical activation, pulsed arterial spin labeling (pASL) scans obtained during a visual task were normalized by either additive or multiplicative normalization of modal gCBF. Normalization by either method increased the statistical significance of cortical activation by a visual stimulus. However, image quality was superior with additive normalization, whether judged by intensity histograms or by reduced variability within gray and white matter.

Corresponding author
Kevin J. Black, kevin@WUSTL.edu

## INTRODUCTION

Blood flow imaging of the brain has brought insights to neuroscience for over 50 years (*Taber, Black & Hurley, 2005*). Whole-brain, or global, cerebral blood flow (gCBF) calculated from blood flow images can be the outcome measure of interest, or can permit correction for gCBF fluctuations that complicate identifying or interpreting relative changes in regional blood flow (rCBF) (*Black et al., 2002*).

Readers more familiar with BOLD (blood oxygen level dependent) imaging must take care not to confuse (a) gCBF calculation and correction with (b) correction for global signal in statistical analysis of BOLD images, a procedure that differs in several important ways. Unlike typical BOLD methods, CBF is a physical attribute with physiologically meaningful values. The typical distribution of CBF across the healthy human brain is well established by a variety of techniques, with a mean of about 50 ml of blood per 100 g of brain tissue per minute (50 ml/hg/min; solid curve in Fig. 1) (*Lassen, 1985*; *Ramsay et al., 1993*). Therefore, any CBF imaging method can be verified by quantitative comparison to

**Peer**J

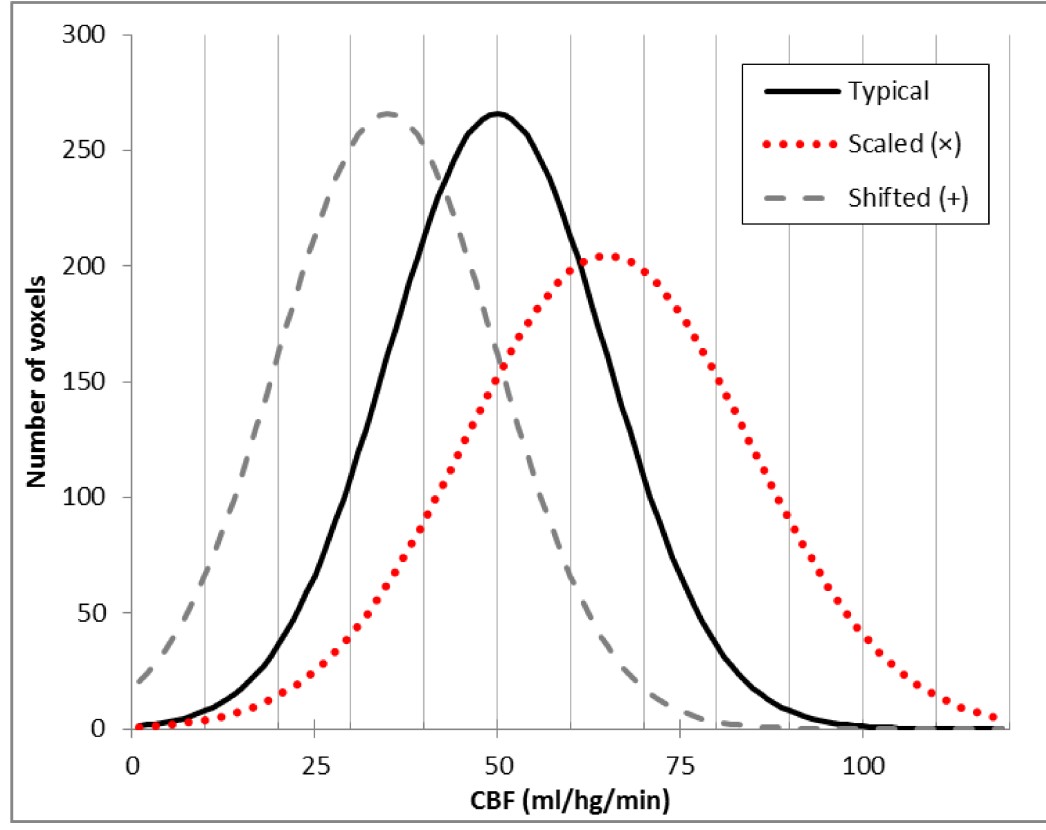

**Figure 1 Typical brain rCBF values and the effects of multiplication or addition.** Histogram of typical rCBF values from resting human brain (solid line). Histograms are also shown for the voxelwise product of that image with 1.3 (dotted line) and for the difference image after subtracting 20 ml/hg/min from each voxel (dashed line).

these standard values. As a consequence, voxels within a single CBF image are expected to demonstrate internally consistent values (e.g., gray matter CBF is similar throughout the brain), again unlike most BOLD methods. Determining global CBF is thus a method aimed primarily at calibrating the voxels within a single CBF image, not a method for accounting for fluctuations at a single voxel across a time series of BOLD images. Another way of saying this is that gCBF computation is aimed at finding a numerical value with physical units, rather than the value of a test statistic.

Early *in vivo* measures of CBF in humans used PET to collect an autoradiographic image of radioactive counts over a defined interval of time (*Herscovitch, Markham & Raichle, 1983*). Measuring arterial radioactivity over time allowed calculation of quantitative rCBF from the PET image of radioactivity concentration (*Herscovitch, Markham & Raichle, 1983*; *Raichle et al., 1983*). Changes in rCBF then could be calculated directly by subtracting two different PET images of rCBF (*Ramsay et al., 1993*). However, arterial sampling required an invasive measurement and added statistical noise. Fortunately, the quantitative CBF image was essentially a scalar multiple of the PET radioactivity concentration image, so when gCBF was not expected to change substantially between two CBF images, one could

multiplicatively scale the raw images to an arbitrary gCBF value before subtraction, and many PET activation studies did just that to account for modest fluctuations of gCBF between PET scans without arterial sampling for quantification (*Fox et al., 1984*).

In 2008 we performed an ASL pharmacological challenge MRI study in Parkinson disease (PD) (*Black et al., 2010b*). The original analysis used multiplicatively scaled images based on our experience with PET blood flow imaging (*Black et al., 2002*). However, when we revisited those data recently for a new analysis (*Stewart et al., 2014*), we sought to improve the quality of the scaled images. Frequency histograms of the perfusion images appeared to differ from the typical rCBF distribution by a fixed shift (dashed line in Fig. 1) rather than by a similar change in mean and dispersion, as one would see with multiplicatively scaled images (dotted line in Fig. 1). Returning to basic principles suggested an explanation. Unlike PET CBF, perfusion images from arterial spin labeling (ASL) MRI are created by subtracting two images obtained a few seconds apart, one in which arterial blood flowing into the brain has been labeled using a spatially limited radio frequency pulse ("tag") and a second image without that label ("control") (*Wang et al., 2003*). Subtraction creates the possibility of a negative or positive additive bias across a CBF image, in which case additive rather than multiplicative correction may better equalize the intensity of two ASL CBF images.

We test that hypothesis here using several criteria to assess image quality.

## MATERIAL & METHODS

### Study participants

Twenty-one nondemented, nondepressed, ambulatory adults age 40–75 with idiopathic PD, treated with a stable dose of levodopa but no dopamine agonists, participated in the study. Detailed inclusion and exclusion criteria and subject characteristics were reported previously (*Black et al., 2010a*; *Black et al., 2010b*). Subjects were enrolled in a Phase 2a dose-finding study (*Black et al., 2010b*), but here we use only data acquired on the placebo day when subjects were in the "practical off state" (i.e., no antiparkinsonian medications for at least 9 h). The study was approved by the Washington University Human Research Protection Office (IRB approval # 08-0059), and all subjects provided written documentation of informed consent prior to participation.

### Subject behavior

Each subject had two perfusion MRI scans while fixating a central crosshair surrounded by a circular checkerboard reversing at 8 Hz, and two control visual fixation scans with the crosshair but no checkerboard.

### MR image acquisition

All MRI data were acquired on a Siemens 3T Tim Trio with matrix head coil. ASL images were acquired with the commercial Siemens pulsed arterial spin labeling (pASL) sequence (*Wang et al., 2003*). Fifteen transverse echo-planar readout slices were acquired with center-to-center slice distance 7.5 mm, $64 \times 64$ voxels in plane with dimensions $(3.4375 \text{ mm})^2$, repetition time (TR) 2.6 s, echo time (TE) 13.0 ms, and flip angle 90°. An

$M_0$ image was followed by 31 tag-control pairs for a total acquisition time for each ASL "scan" of 2.73 min.

Brain structure was assessed from sagittal MP-RAGE acquisitions with voxel size $(1.0 \text{ mm})^3$, TR = 2.4 s, TE = 3.08 ms, TI = 1,000 ms, flip angle = 8°. The structural images for each subject were inspected visually, images of lower quality were rejected, and the remaining 1–4 MP–RAGE images for each subject were mutually registered.

## Image registration and creation of CBF images

The 63 frames of the ASL run were rigidly aligned using a validated method (*Black et al., 2001*) and summed to facilitate later alignment steps. Each frame was smoothed using a Gaussian filter with 7.35 mm kernel (FWHM), and cerebral blood flow (CBF) was computed in each voxel for each tag-control pair as described (*Wang et al., 2003*). The summed, aligned EPI images from each run were mutually aligned within each subject and summed across runs, and the resulting image was affine registered to a target image in Talairach and Tournoux space made using validated methods from the structural MR images from these subjects (*Black et al., 2004*). The products of the registration matrix from this step and the matrices from the within-run mutual registration step were used to resample the 31 tag-control pair CBF images from each run into atlas space in a single resampling step. A whole-brain binary image was created from the atlas-space structural target image, and all analyses below included only the CBF image voxels corresponding to voxels within this mask image.

To minimize motion-related artifact we removed tag–control pairs from all further analyses if framewise displacement in either EPI image exceeded 0.9 mm as defined by *Siegel et al. (2014)*. The remaining CBF images in atlas space were averaged to create a single atlas-registered CBF image for each ASL run. One subject's data was excluded from further analysis because over half of the frame pairs were removed due to head motion.

## Image intensity correction

### Estimating modal CBF image intensity

We evaluated each CBF image independently for each of the 4 scans in each subject. The image intensity histograms were constructed with bins 1 unit wide, so were not smooth. We chose to normalize image intensity based on the idealized peak of this distribution (which if there were no noise would be the mode, i.e., the most common voxel intensity in the brain) (*Ojemann et al., 1997*). Specifically, the method of least squares was used to identify the vertex of the parabola that best fit the histogram using voxels within 70% of the mode bin (Fig. 2).

### Additive and multiplicative intensity correction

Each input image was corrected in two ways: multiplicatively (multiplying every voxel in the image by 50/*mode*), and additively (adding 50-*mode* to every voxel), so that the idealized modal CBF for every corrected image was 50 (nominal units mL/hg/min).

**Figure 2 Determining the idealized mode.** The figure shows the frequency histogram for one of the CBF images. With this bin width, the true mode falls at 32 (frequency = 898, bin including all voxels with values 31.5–32.5), but the mean of the distribution and the vertex of the parabola shown (the idealized mode) both fall at 35.0.

## Defining volumes of interest

Gray matter (GM), white matter (WM) and visual cortex volumes of interest (VOIs), excluding CSF regions, were defined from each subject's high-resolution MP-RAGE image by FreeSurfer (version 5.3, http://surfer.nmr.mgh.harvard.edu) (*Desikan et al., 2006*). VOIs were limited to voxels that were represented in every image in every subject; this step excluded much of the inferior occipital cortex in the visual cortex VOI (*Black et al., 2010b*). We also used the thalamic VOIs previously defined in our drug study (*Black et al., 2010b*).

## Statistical analysis

To determine the effect of gCBF normalization on task effect, all statistical analyses were performed in triplicate, one for each set of images: uncorrected (before removing the gCBF effect), multiplicatively normalized, and additively corrected.

### *Image quality analysis*

Using only the fixation scans, the mean rCBF and its variability in the GM, thalamic and WM VOIs was determined. The gray to white matter CBF ratio was also calculated.

### VOI analysis

To determine the change in rCBF in a VOI with visual stimulation, the mean CBF across all voxels in the VOI from the fixation scans was subtracted from the corresponding mean in the task scans. Statistical significance was assessed with paired $t$ tests.

### Statistical images

To identify regions activated by the visual task, we used a mixed-effects approach. First, for each study subject, changes in rCBF were identified using SPM12b software (www.fil.ion. ucl.ac.uk/spm/) and a voxelwise general linear model that included task (checkerboard) versus control (fixation). The task contrast was used to generate an image of the weighted parameter estimate for each subject. These single-subject contrast images were used as input for a second SPM analysis using a voxelwise general linear model that included a covariate for subject age and a factor for sex. Statistical inference was performed at each voxel with a one-sample $t$ test (i.e., testing whether the task contrast images are significantly less than or greater than zero, across subjects). After thresholding at $p = 0.001$, multiple comparisons correction was performed with the cluster false discovery rate set at $p = 0.05$. Approximate anatomical locations of peaks in the statistical images were provided by the Talairach Daemon client (www.talairach.org) (*Lancaster et al., 2000*).

## RESULTS

We assessed the effect of additive or multiplicative intensity equalization in several ways: by examining the voxel intensity distribution (by frequency histogram), by judging image quality visually, by the variability of voxel intensity within the GM and WM, and by the suitability of each set of images for detecting appropriate brain activation with visual stimulation.

### Voxel intensity distribution (histogram)

The idealized mode $\pm$ SD for the original CBF images was $33.36 \pm 7.30$. The original CBF images contained a reasonable distribution of voxel intensities except that many of them appeared shifted leftwards to varying degrees, so that many voxels in the brain had physiologically impossible negative values (the 3 curves in the histogram in Fig. 3A reflect all brain voxels from 3 successive ASL images from the same subject; a transverse section from each of these images is shown to the right of the histogram). Multiplicative normalization of course produced an image with an equal fraction of negative voxel values, though the normalized image's mode was now 50 (histograms in Fig. 3B). Additive normalization produced a voxel intensity distribution that reflects the physiological expectation. In addition, it much more accurately matches the intensity distribution widths across the normalized images (histograms in Fig. 3C).

### Image quality

Images normalized additively (Fig. 3C) appear to show more uniform voxels within the GM and WM. To corroborate this quantitatively, we determined the variability in the GM and WM both between subjects and within subjects, using scans from the fixation task only.

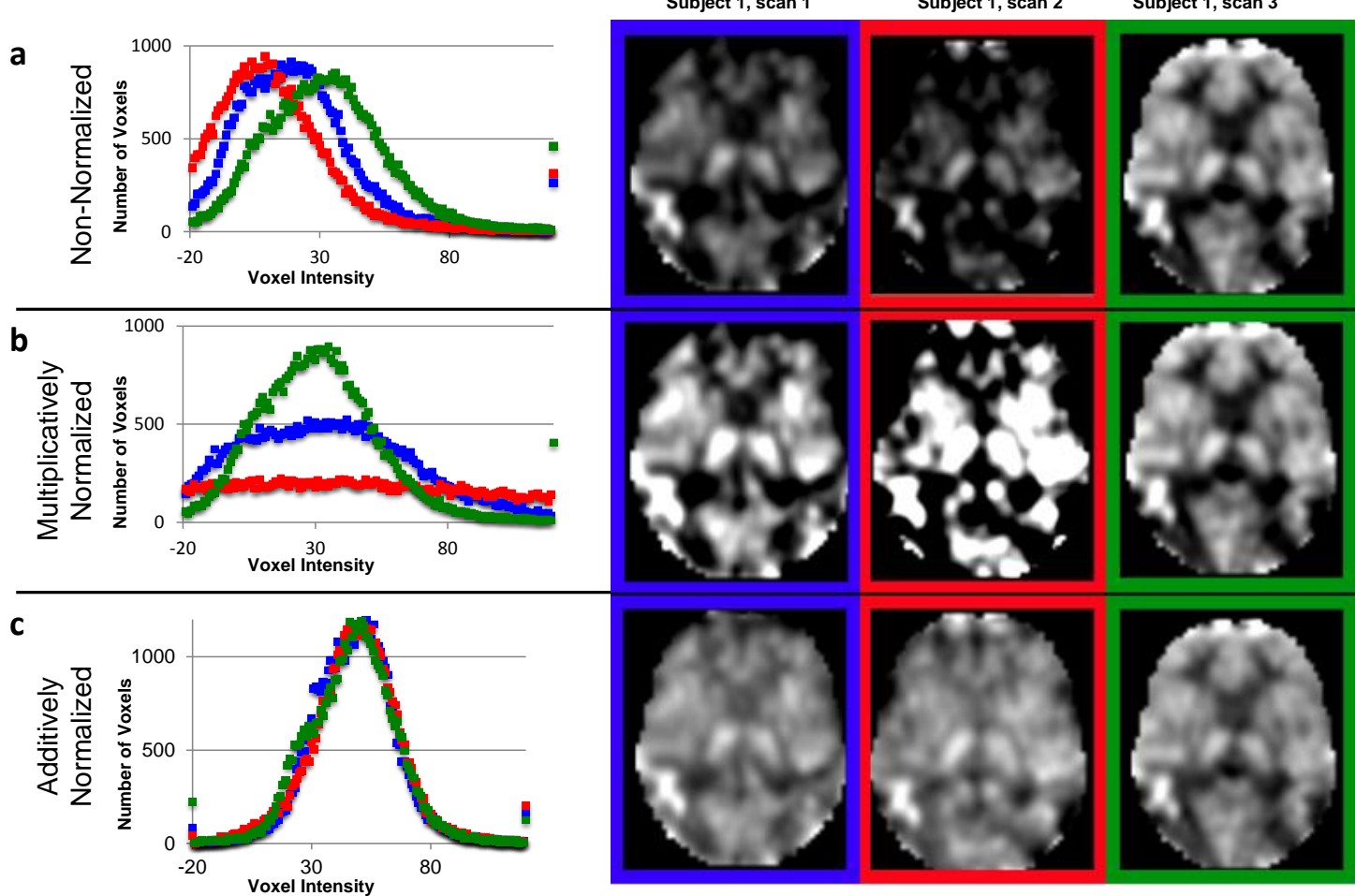

**Figure 3 Additive intensity normalization improves pASL image quality.** The effect of multiplicative or additive intensity normalization on 3 ASL scans (represented by red, blue and green) from one single subject: (A) before normalization, (B) after multiplicative normalization, (C) after additive normalization. The histograms show the frequency of the voxel values across the brain (i.e., within an anatomically-defined brain mask only) for each of 3 sequential scans. The same transverse slice in atlas space from the 3 sequential scans is also shown; scan 1 (blue), scan 2 (red) and scan 3 (green). The grayscale range is set at 0–100 ml/hg/min, a range that reasonably reflects true physiological values, except for the images before normalization (A), for which the range is set at 0–75 ml/hg/min because otherwise the structure in some of the images would be obscured.

Across subjects, the mean GM intensities are more similar in additively normalized scans (coefficient of variation [CV] = 0.145) as compared to non-normalized (CV = 0.194) or multiplicatively normalized scans (CV = 0.266). Within the GM, the thalamic VOI also had mean intensities more similar in additively normalized scans (coefficient of variation [CV] = 0.0093) as compared to non-normalized (CV = 0.154) or multiplicatively normalized scans (CV = 0.167). The WM intensities are more similar in additively normalized scans (CV =0.158; non-normalized CV = 0.206, multiplicatively normalized scans CV = 0.263). The GM:WM ratio is more similar across subjects after additive normalization (CV =0.056; non-normalized CV = 0.074, multiplicatively normalized CV = 0.072) (Table 1).

**Table 1  Variability across subjects of mean GM, thalamus and WM rCBF.**

| | Non-normalized images | Multiplicatively normalized | Additively normalized |
|---|---|---|---|
| Mean GM rCBF (SD, CV)[a] | 38.78 (7.51, 0.194) | 62.02 (16.47, 0.266) | 55.89 (8.11, 0.145) |
| Mean Thalamus rCBF (SD, CV)[a] | 54.80 (8.42, 0.154) | 84.66 (14.10, 0.167) | 71.41 (6.67, 0.093) |
| Mean WM rCBF (SD, CV)[a] | 27.56 (5.68, 0.206) | 43.85 (11.54, 0.263) | 44.67 (7.06, 0.159) |
| Mean GM:WM ratio (SD, CV) | 1.416 (0.104, 0.074) | 1.418 (0.102, 0.072) | 1.256 (0.070, 0.056) |

Notes.
[a] Nominal units ml/hg/min

**Table 2  rCBF changes in visual cortex and WM VOIs with visual stimulation.**

| | Non-normalized images | Multiplicatively normalized | Additively normalized |
|---|---|---|---|
| Mean change in rCBF (SD, CV) in visual cortex region[a] | 17.70 (13.88, 0.784) | 29.83 (18.91, 0.634) | 18.15 (13.37, 0.737) |
| $p$ | $1.70 \times 10^{-5}$ | $1.03 \times 10^{-6}$ | $7.72 \times 10^{-6}$ |
| Mean change in rCBF (SD, CV) in white matter region[a] | 1.07 (3.23, 3.01 ) | 2.85 (5.99, 2.10) | 1.53 (3.47, 2.28) |
| $p$ | 0.15 | 0.05 | 0.06 |

Notes.
[a] Nominal units ml/hg/min. Note that the mean changes are not strictly comparable between methods because of the multiplication. Hence we provide SD and $p$ values for comparison.

## Task activation: a priori volume of interest

We examined the effect of visual stimulation in the partial visual cortex VOI (intended as a positive control) and in the WM VOI (intended as a negative control). With this substantial visual stimulus, the signal is detected even without normalization (visual cortex VOI, $p = 1.70 \times 10^{-5}$, paired $t$ test), but the statistical significance of the change is greater in either set of normalized images (multiplicative normalization, $p = 1.03 \times 10^{-6}$, additive normalization, $p = 7.72 \times 10^{-6}$). The mean change in rCBF in the visual cortex VOI increased when the images were multiplied, but was always >10 times higher than the change in the WM control VOI (Table 2).

## Task activation: statistical image

Similarly to the VOI analysis, with this substantial visual stimulus, SPM analyses of visual stimulation identified significant occipital cortex activation even without normalization, though normalization increased the peak $t$ and the volume of significant clusters (Fig. 4 and Table 3). For all 3 analyses, the peak $t$ value occurred in the occipital lobe, in Brodmann area 17 or 18. There were no significant clusters of deactivation with any of the analyses.

## DISCUSSION

Additive normalization of gCBF is superior to multiplicative normalization for this pASL technique, judged by image intensity distributions and by homogeneity within gray matter

Non-normalized

Multiplicatively Normalized

Additively Normalized

**Figure 4  SPM regions of activation.** Top row: before normalization with (A) sagittal, (B) coronal and (C) axial slices; middle row: after multiplicative normalization (D) sagittal, (E) coronal and (F) axial slices; bottom row: after additive normalization with (G) sagittal, (H) coronal and (I) axial slices. The lower half of occipital cortex was excluded from this analysis due to incomplete superior-inferior coverage in some subjects (see Methods: Defining volumes of interest).

and white matter. Multiplicative scaling increases the variability in GM and WM, whereas additive normalization improves image quality and reduces variability between subjects.

We previously found that correcting for gCBF improved task signal in ASL in young, healthy volunteers (*Black et al., 2008*; *Henniger et al., 2009*). The present report extends those findings to patients with Parkinson disease. Normalizing scans for variability in gCBF improves detection of statistically significant activations from a visual task, whether analyzed using an anatomically-defined visual cortex VOI or by whole-brain SPM. Although noise properties improve to a greater extent with additive scaling, either

**Table 3 Activation by a visual stimulus: SPM analyses.**

| | Non-normalized images | Multiplicatively normalized | Additively normalized |
|---|---|---|---|
| Peak $t$ (17 d.f.) | 6.13 | 7.36 | 6.48 |
| Location of peak | −8, −93, 6 | 4, −81, 0 | −8, −93, 6 |
| Cluster volume, voxels (ml) | 119 (3.2) | 369 (10.0) | 447 (12.1) |
| $p$ (FDR) | <0.0005 | <0.0005 | <0.0005 |
| Peak $t$ for 2nd significant cluster | 5.59 | | |
| Location of peak | 26, −87, 9 | | |
| Cluster volume, voxels (ml) | 247 (6.7) | | |
| $p$ (FDR) | <0.0005 | | |

additive or multiplicative scaling permits identification of rCBF responses to a strong visual stimulus.

The reader familiar with ASL should note a different and potentially confusing use of the term "global." Before calculating CBF, several pre-processing quality improvement steps are applied to the EPI images to address signal artifacts caused by head movement or MR signal drift over time within an ASL run. *Wang (2012)* used a general linear model and global EPI signal intensity (defined by voxels above an arbitrary intensity threshold) to account for such artifactual signal changes. We chose to address these issues using methods developed and validated at our institution; for instance, we removed motion-contaminated frames (*Siegel et al., 2014*) rather than modeling the consequent artifact. Again, please note that Wang's within-run global signal variable, applied separately at each voxel, is quite different from the global (whole-brain) single-scan measurement of CBF compared between runs in this communication.

### Funding

The data came from a study funded by Synosia Therapeutics. The analysis reported here was supported by NIH (K24 MH087913, T32 DA007261, NS058714, NS41509, NS075321, C06 RR020092, UL1 RR024992, P30 NS048056, U54 CA136398-02900209); the American Parkinson Disease Association (APDA) Center for Advanced PD Research at Washington University, the Greater St. Louis Chapter of the APDA, and the Barnes-Jewish Hospital Foundation (Elliot Stein Family Fund for PD Research and the Parkinson Disease Research Fund). The original study was funded commercially, but the sponsor did not participate in or affect this analysis or this report. There have been no other commercial or financial relationships that could be construed as a potential conflict of interest. The funders had no role in data analysis, decision to publish, or preparation of the manuscript.

### Grant Disclosures

The following grant information was disclosed by the authors:
Synosia Therapeutics.
NIH: K24 MH087913, T32 DA007261, NS058714, NS41509, NS075321, C06 RR020092, UL1 RR024992, P30 NS048056, U54 CA136398-02900209.
American Parkinson Disease Association.
Center for Advanced PD Research at Washington University.
Greater St. Louis Chapter of the APDA.
Barnes-Jewish Hospital Foundation.

### Competing Interests

The original study was funded commercially, but the sponsor did not participate in or affect this analysis or this report. There have been no other commercial or financial relationships that could be construed as a potential conflict of interest. Kevin Black, MD is an Academic Editor for PeerJ.

### Author Contributions

- Stephanie B. Stewart analyzed the data, wrote the paper, prepared figures and/or tables.
- Jonathan M. Koller and Meghan C. Campbell conceived and designed the experiments, performed the experiments, analyzed the data, reviewed drafts of the paper.
- Joel S. Perlmutter wrote the paper.
- Kevin J. Black conceived and designed the experiments, performed the experiments, analyzed the data, wrote the paper.

### Human Ethics

The following information was supplied relating to ethical approvals (i.e., approving body and any reference numbers):

Washington University Human Research Protection Office, approval # 08-0059.

### Supplemental Information

Supplemental information for this article can be found online at http://dx.doi.org/10.7717/peerj.834#supplemental-information.

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
