# Peer review of "Additive global cerebral blood flow normalization in arterial spin labeling perfusion imaging"

_PeerJ, doi:10.7717/peerj.834_

## Round 0.1 · original submission · Major Revisions

The most significant requested revision is the need to redo the analysis with contrast or beta images rather than t-images (unless this is what was actually done but incorrectly reported). This may substantially alter the results.

Reviewer 1 ·

Basic reporting

The text could be polished more to improve readability by a wider audience. Additional labeling could be added to enhance the visualization of Figures 3-4.

Experimental design

VOI analyses were somewhat limited by restricting to GM and WM only given that CBF was not homogeneous in these very large regions. This part of the findings would be more interesting if the authors report the analogous results in subdivisions of GM and WM across the two hemispheres (e.g., different lobes or cortico-subcortical brain regions relevant to Parkinson's disease.

Validity of the findings

In addition to the use of the histograms to illustrate the problem and the solution the authors compared the variability in large GM/WM regions and the sensitivity of detecting regional activation with SPM analysis - without normalization and with normalization by multiplicative and additive methods. The resulting work ensured the validity of the results in this first study of such a problem.

Additional comments

This study described a novel method for normalizing global effects of CBF images acquired by perfusion MRI with arterial spin labeling (ASL). The work was performed to solve the potential problem caused by negative values resulting from the subtraction procedure involved in preprocessing ASL scans. By using activation data in patients with Parkinson’s disease before and after visual stimulations, the authors demonstrated the superiority of additive normalization as compared to the conventional technique of ratio normalization. This method may be very useful in enhancing signal-to-noise characteristics of ASL MRI in regional or voxelwise brain mapping analysis. However, further validation is still necessary to evaluate the effectiveness of this new approach in neuroimaging research.

Minor points:
Line 127: state clearly whether CSF was excluded in the brain mask which usually includes gray matter and white matter.
Line 149: specify whether VOIs were created in native space of the subject or in standard Talairach and Tournoux space.
It appeared that Black et al 2002a and Black et al 2002b referred to the same article – either update or correct this minor issue.
Fig 3 – label subject 1, 2, 3 on top of the three image columns – to make it directly visible.
Fig 4 – label each panel to identify what it is – so that the reader can see it explicitly.
Tab 2 – add the values of CV as in Tab 1.

Reviewer 2 ·

Basic reporting

This manuscript explores how multiplicative and additive global normalization affect PASL image quality, as well as statistical significance of activations in response to a visual task. This is an important article for the neuroimaging community as ASL has seen a rapid increase in applications in recent years and because ASL images have very different characteristics compared to traditional BOLD images, optimal data analysis techniques from the much larger body of BOLD literature are not readily transferrable to ASL. On the whole, the paper is well-structured with sound results and conclusions. However, I had a few questions and comments regarding the methods section:

1. Each subject has 4 scans: 2 active, 2 rest, each with 31 difference images. Are these normalized by the modal CBF of each scan or the average of all 4?

2. In the first level analysis, are these 31x4 timeframes modeled as a block design with 2.73min blocks?

3. In the statistical images section, the authors stated they used the t-images are input for second-level analysis, which is not appropriate. Please rerun the analysis with the contrast images.

4. Early studies on the effects of proportional scaling in fMRI analysis often report artifactual deactivations. Have the authors looked at deactivations across the different normalization techniques used in this manuscript? It may provide further support for one method or the other.

Also, a minor correction in line 41: remove “correction with”

Experimental design

no comments

Validity of the findings

no comments

---

## Round 0.2 · accepted · Accept

I have no additional Comments.

Reviewer 1 ·

Basic reporting

No comments

Experimental design

No comments

Validity of the findings

The results have been strengthed with the changes in the revision.

Additional comments

The authors have dealt with the major issues identified by this reviewer. The paper is much improved from the last version.

Reviewer 2 ·

Basic reporting

The authors have addressed all of my concerns in the initial review.

Experimental design

No comment

Validity of the findings

No comment